# Decreased Expression of Plakophilin-2 and αT-Catenin in Arrhythmogenic Right Ventricular Cardiomyopathy: Potential Markers for Diagnosis

**DOI:** 10.3390/ijms23105529

**Published:** 2022-05-16

**Authors:** Pei-Fang Hung, Fa-Po Chung, Chung-Lieh Hung, Yenn-Jiang Lin, Tzu-Ting Kuo, Jo-Nan Liao, Yun-Yu Chen, Chih-Hsin Pan, Kai-Ping Shaw, Shih-Ann Chen

**Affiliations:** 1Heart Rhythm Center and Division of Cardiology, Department of Medicine, Taipei Veterans General Hospital, Taipei 112201, Taiwan; berthepei@gmail.com (P.-F.H.); linyennjiang@gmail.com (Y.-J.L.); jnliao@vghtpe.gov.tw (J.-N.L.); r01847021@gmail.com (Y.-Y.C.); sachen@vghtpe.gov.tw (S.-A.C.); 2Institute of Clinical Medicine and Cardiovascular Research Center, National Yang Ming Chiao Tung University, Taipei 112304, Taiwan; absintholic@hotmail.com; 3Department of Medicine, Mackay Medical College, New Taipei City 252005, Taiwan; jotaro3791@gmail.com; 4Institute of Biomedical Sciences, Mackay Medical College, New Taipei City 252005, Taiwan; 5Division of Cardiology, Department of Internal Medicine, Mackay Memorial Hospital, Taipei 104217, Taiwan; 6Division of Cardiovascular Surgery, Department of Surgery, Taipei Veterans General Hospital, Taipei 112201, Taiwan; 7Institute of Epidemiology and Preventive Medicine College of Public Health, National Taiwan University, Taipei 100025, Taiwan; 8Institute of Forensic Medicine, Ministry of Justice, New Taipei City 235016, Taiwan; acpanp501361@gmail.com (C.-H.P.); kpshaw596@gmail.com (K.-P.S.)

**Keywords:** arrhythmogenic right ventricular cardiomyopathy, CTNNA3, αT-catenin, plakophilin-2, immunohistochemistry staining

## Abstract

Arrhythmogenic right ventricular cardiomyopathy (ARVC) is a hereditary disease of the heart muscle. Clinical challenges remain, however, in identifying patients with ARVC in the early or concealed stages with subtle clinical manifestations. Therefore, we wanted to identify potential targets by immunohistochemical (IHC) analysis in comparison with controls. Pathogenic mutations were identified in 11 of 37 autopsied patients with ARVC. As observed from IHC analysis of the RV, expression of αT-catenin and plakophilin-2 is significantly decreased in autopsied patients with ARVC as compared to controls, and the decreased expression is consistent in patients with and without pathogenic mutations. Furthermore, ARVC specimens demonstrated a reduced localization of αT-catenin, desmocollin-2, desmoglein-2, desmoplakin, and plakophilin-2 on intercalated discs. These findings have been validated by comparing RV specimens obtained via endomyocardial biopsy between patients with ARVC and those without. The pathogenic mutation was present in 3 of 5 clinical patients with ARVC. In HL-1 myocytes, siRNA was used to knockdown CTNNA3, and western blotting analysis demonstrated that the decline in αT-catenin expression was accompanied by a significant decline in the expression of plakophilin-2. The aforementioned effect was directed towards protein degradation rather than mRNA stability. Plakophilin-2 expression decreases concurrently with the decline in CTNNA3 expression. Therefore, the expression of αT-catenin and plakophilin-2 could be potential surrogates for the diagnosis of ARVC.

## 1. Introduction

Arrhythmogenic right ventricular cardiomyopathy (ARVC) is a hereditary disorder of the heart muscle. Pathologically, ARVC is characterized by progressive fibro-fatty replacement of the right ventricle (RV) [1,2]. A primary cause of ARVC is the disruption of cell–cell adhesion, in which cardiac intercalated discs (ICDs) play a pivotal role. Up to 60% of ARVC patients have mutations in genes that encode desmosomal proteins. Moreover, mutations in several genes encoding non-desmosomal proteins have been associated with the pathogenesis of ARVC [3,4,5,6,7,8,9,10,11,12,13,14,15]. Currently, the modified Task Force Criteria have been proposed to facilitate the diagnosis of ARVC [16]. Nevertheless, it remains difficult to identify patients with ARVC in the early or concealed stages with subtle clinical manifestations.

By immunohistochemical analysis of endomyocardial biopsy samples, it has previously been demonstrated that a decreased expression of plakoglobin in the ICDs and connexin 43 could be identified in ARVC [17], which can assist in the determination of a clinical diagnosis. On the other hand, quantitative IHC staining could reveal an insignificant decrease in junction protein expression when comparing autopsies from patients with ARVC to controls [18]. αT-catenin interacts with N-cadherins and desmosomal cadherins, particularly plakophilin-2, suggesting an important molecular association between adherens junctions and desmosomes in the heart [19]. However, the protein stability of the Plakophilin-2 is known to be regulated by cysteine proteases [20]. Additionally, the aforementioned findings were supported by the fact that CTNNA3 is a causative gene for ARVC [21]. Nevertheless, the effects of CTNNA3 on the expression of desmosomal proteins and αT-catenin in ARVC have not been fully elucidated.

The purpose of the present study was to explore the expression of αT-catenin and desmosomal-associated proteins in RV tissues collected from autopsied participants and clinical patients with ARVC using IHC staining. The transient knockdown of CTNNA3 using cultured cardiomyocytes was used to validate the abovementioned findings by western blotting.

## 2. Materials and Methods

### 2.1. Collection of Right Ventricle Tissue Samples

First, 48 RV specimens were collected from 36 autopsied patients with ARVC and 12 controls autopsied for non-cardiac causes from the Taiwan National Forensic Institute database. Secondly, clinical RV specimens were collected via endomyocardial biopsy from eight and four patients with and without definite ARVC, respectively. Those samples were then stained by IHC. The Institute of Forensic Medicine and Taipei Veterans General Hospital reviewed and approved this study.

### 2.2. Diagnosis of ARVC

Among 36 autopsied patients with ARVC from the Taiwan National Forensic Institute registry, the diagnosis was confirmed based on gross and histopathological findings, including extensive fibro-fatty replacement of the RV [22,23,24]. As described in a previous study, this condition was characterized by substitution of the myocardium with sparing of the trabecular myocardium [25]. In brief, transmural myocardial fatty infiltration usually involves the posterobasal, anteroapical, and infundibular regions or the entire RV. Aneurysms of the RV characterized by thinning of the free wall or bulging of the wall have been described previously [26]. Patients with fibro-fatty infiltration confined to the anteroapical region alone, which was considered a normal finding, were excluded from the study [27]. The diagnosis of ARVC was independently confirmed by two forensic pathologists.

A clinical diagnosis of ARVC is made according to the 2010 revised Task Force Criteria [16]. None of the controls met the criteria for either borderline or possible ARVC.

### 2.3. Genetic Testing

Genomes of heart tissues or blood samples collected from autopsied participants with ARVC and controls were sequenced by targeted next-generation sequencing on the Illumina Hi-seq2000 platform (Illumina, San Diego, CA, USA). These genomes contained PKP2, DSC2, DSG2, DSP, JUP, DES, TMEM43, TGFβ3, RYR2, CTNNA3, PKP4, and PLN according to the ARVC genetic variants database. Based on the American College of Medical Genetics and Genomics guidelines, we filtered and evaluated each variant for pathogenicity and identified pathogenic, likely pathogenic, and variants of uncertain significance [28].

### 2.4. Immunohistochemistry

A 4-μm-thick section of right ventricular tissue, embedded in paraffin and deparaffinized with Trilogy Solution (Cell Marque Corp., Rocklin, CA, USA) at 121 °C for 10 min, was obtained from autopsied patients or patients with endomyocardial biopsy. The sections were treated with 3% H_2_O_2_ methanol and then incubated with DakoCytomation Dual Endogenous Enzyme Block (DakoCytomation Inc., Carpinteria, CA, USA) for 10 min; Ultra V Block (Lab Vision Corporation, Fremont, CA, USA) for 10 min; antibody dilution buffer (Ventana Medical Systems Inc., Oro Valley, AZ, USA) for 10 min; and antibodies against αT-catenin, plakophilin-2 (Thermo Fisher Scientific, Waltham, MA, USA), desmin (eBioscience, San Diego, CA, USA), desmocollin-2 (Cusabio Biotech Co., Houston, TX, USA), desmoglein-2, ryanodine receptor 2, transmembrane protein 43, transforming growth factor-beta 3, tenascin N (Abcam, Cambridge, UK), desmoplakin (Santa Cruz Biotechnology, Dallas, TX, USA), and plakoglobin (Sigma-Aldrich, St. Louis, MO, UK) overnight at 4 °C. In accordance with the manufacturer’s instructions, immunoreactivity was assessed using the BioGenex Super Sensitive Link-Label IHC Detection System (BioGenex, Fremont, CA, USA). We randomly selected and evaluated ten regions of the RV tissues. ImageJ was used to quantify the density of immunoreactivity in the regions of each protein.

### 2.5. Cell Lines and Culture Conditions

The HL-1 cells were derived from mouse atrial cardiomyocytes and cultured on Corning CellBIND surface culture dishes (Corning, Glendale, AZ, USA) in Claycomb medium (Sigma-Aldrich) supplemented with 10% fetal bovine serum, penicillin/streptomycin/amphotericin B (Invitrogen, Carlsbad, CA, USA), 0.1 mM norepinephrine, and 2 mM L-glutamine (Sigma-Aldrich) at 37 °C in a humidified atmosphere containing 5% O_2_.

We placed the cells in 60-mm dishes and incubated them at 37 °C overnight (for drug treatment). MG101 (MedChemExpress, Monmouth Junction, NJ, USA) was diluted to a working concentration of 5 µM in the medium. The cells were also treated with an equal volume of DMSO (Sigma-Aldrich) as a control. The cell lysate was harvested after 24 h of treatment and analyzed by gel electrophoresis and western blotting.

### 2.6. Small Interfering RNAs against CTNNA3

ON-TARGETplus Non-targeting Pool siRNA was obtained from Dharmacon and used as the control in this study. The pool contains four small interfering RNAs (siRNAs). The target sequences were as follows: UGGUUUACAUGUCGACUAA, UGGUUUACAUGUUGUGUGA, UGGUUUACAUGUUUUCUGA, and UGGUUUACAUGUUUUCCUA. On-TARGETplus Mouse CTNNA3 siRNA was used against CTNNA3 in HL-1 cells, and the target sequences were as follows: GGAUGAAGGCUCGGCUAA, ACGGAGUACAUGAGUAAUA, CCGCAAGAGUAGCUCACAU, and GGGAGAACUCAUCGUAUCA.

The siRNA transfections were performed using Lipofectamine 3000 (Invitrogen) according to the manufacturer’s recommended protocol. In the RNA silencing assay, 6 × 10^5^ HL-1 cells were plated in six-well plates and incubated at 37 °C overnight. Transfection complexes (containing Lipofectamine 3000 and 75 pmol siRNA) were added directly to the medium. Protein samples were collected 48 h after transfection.

### 2.7. Lentiviral Transduction

Following the establishment of stable cell lines, 5 × 10^5^ HL-1 cells were plated in 60-mm plates and incubated overnight at 37 °C. The cells were transduced using shLuc, shCTNNA3, or shPKP2 lentivirus from RNAi core (Academia Sinica, Taipei, Taiwan) with polybrene (Sigma), and then selected after treating with 2 μg/mL puromycin.

### 2.8. Western Blotting Analysis

Western blot analysis was used to quantify the expression levels of the selected proteins in HL-1 myocytes. Proteins were extracted from cells using a RIPA lysis buffer containing 25 mM tris hydrochloride (pH 7.6), 150 mM sodium chloride, 1% sodium deoxycholate, 0.1% sodium dodecyl sulfate, and protease inhibitor (Roche Diagnostics, Mannheim, Germany). The samples were homogenized by sonication and the cell debris was removed by centrifugation at 13,000 rpm for 30 min at 4 °C. We measured the protein concentration using a BCA protein quantitative analysis kit (Perkin Elmer Corporation, Waltham, MA, USA).

Aliquots of 30 μg of protein extracts were denatured by boiling at 100 °C for 10 min in 5× Laemmli buffer (Bio-Rad Laboratories, Hercules, CA, USA). Denatured proteins were separated into 8–12% sodium dodecyl sulfate-polyacrylamide resolving gel, and then transferred to pre-equilibrated polyvinylidene fluoride membranes (Perkin Elmer Corporation). The membranes were blocked with 5% nonfat milk in TBST for 1 h at room temperature and then incubated with antibodies against desmosomal-associated proteins, which include: αT-catenin, connexin 43 (Sigma-Aldrich), desmocollin-2, desmoglein-2, desmoplakin, plakophilin-2 (Abcam), and plakoglobin (BD Transduction Laboratories, Franklin Lakes, NJ, USA); fibrosis markers: transforming growth factor-beta 3 (Abcam); and internal control: glyceraldehyde-3-phosphate dehydrogenase (GAPDH; Ambion, Austin, TX, USA) overnight at 4 °C. Membranes were washed with TBST 3 times for 10 min and then incubated with HRP-conjugated secondary antibody (DakoCytomation Inc.) at room temperature for 1 h. The immunoreactive proteins were visualized with thechemiluminescent reagent ECL (Millipore, Burlington, MA, USA). We used VisionWorks 8.0 software to perform a densitometric analysis of the protein expression level.

### 2.9. RNA Extraction and Real-Time Quantitative Polymerase Chain Reaction (PCR)

RNA was isolated from HL-1 cardiomyocytes using TRIzol reagent (Invitrogen) according to the manufacturer’s instructions. The total RNA was reverse-transcribed for 30 min at 50 °C using the UltraScript 2.0 cDNA Synthesis Kit (PCR Biosystems Ltd., London, UK) and random hexamer primers. Moreover, the mouse CTNNA3 and PKP2 transcript levels were analyzed using a QuantStudio™ 3 Real-Time PCR System (Thermo Fisher Scientific) with Power SYBR Green PCR Master Mix (Thermo Fisher Scientific) and the following primer sequences: CTNNA3 forward, 5′-TGAGATTGAGATATGGGATG-3′, and reverse, 5′-CAGTGGTATGCTTTAGTGGT-3′; PKP2 forward, 5′-CAGGTGCTGAAGCAAACCAGAG-3′, and reverse, 5′-GACACTCTCTGTCAAGGTGAGC-3′; and β-actin forward, 5′-GGCTGTATTCCCCTCCATCG-3′, and reverse, 5′-CCAGTTGGTAACAATGCCATGT-3′. β-actin was used as the internal control.

### 2.10. Statistical Analysis

The categorical variables were expressed as numbers and proportions and were compared using either chi-square (χ2) test or Fisher’s exact test. The continuous variables were expressed as mean ± standard deviation. The differences in the quantitative results of IHC staining between the two groups were compared using the Wilcoxon signed-rank test (Mann–Whitney U test). The Statistical Package for the Social Sciences software 22.0 (IBM Corporation, Armonk, NY, USA) was used in all analyses. A *p*-value < 0.05 was considered statistically significant.

## 3. Results

### 3.1. Specimens Collection

From 2012 to 2017, we collected RV tissue samples from 49 Taiwanese (38 men, age: 35.61 ± 12.80 years) autopsied at Taiwan National Forensic Institute. Among them, 37 were diagnosed with ARVC based on histopathological findings, and 12 controls died from noncardiac causes. Furthermore, no structural abnormalities could be identified in the controls. The ARVC participants were younger than the controls (32.86 ± 10.30 vs. 44.17 ± 16.45 years, *p* = 0.021, Table 1).

### 3.2. Immunohistochemical Staining of Desmosomal-Associated Proteins between ARVC and Non-ARVC Samples

Figure 1a shows the results of the IHC. Accordingly, Table 1 illustrates the quantitative analysis results of immunoreactivity of each protein. It is noteworthy that the expression of αT-catenin and plakophilin-2 was significantly lower in the ARVC group compared to the control group (αT-catenin: 51.03 ± 9.79 vs. 71.63 ± 20.56, *p* < 0.001; plakophilin-2: 62.40 ± 12.91 vs. 71.00 ± 10.84, *p* = 0.034; Table 1). The levels of protein expression were not significantly different between the control and ARVC groups.

### 3.3. Immunohistochemical Staining of ARVC Specimens with and without Identified Gene Mutations

The pathologic gene mutations have been identified in 11 (29.73%) of 37 autopsied ARVC participants. Among these patients, one (9.09%) and ten (90.91%) had multiple gene mutations (DSP and CTNNA3) and single-gene mutations, respectively. Appendix A Table A1 demonstrates detailed information regarding the specimens with identifiable mutations.

The IHC results are shown in Table 2. According to the analysis, there were no significant differences in terms of protein expression between specimens with and without identifiable mutations. The results suggest that a decreased expression of αT-catenin and plakophilin-2 could be considered a surrogate marker of ARVC.

Figure 1b showed the relative protein expression patterns in representative specimens collected from autopsied participants. In comparison with controls, the levels of αT-catenin and plakophilin-2 were almost lower in those with and without identifiable gene mutations in ARVC. This result indicated the potential role of IHC staining of αT-catenin and plakophilin-2 in the diagnosis of ARVC.

As shown in Figure 1c, the area under the receiver operating characteristic curves of αT-catenin, plakophilin-2, and both proteins were 0.812 (95% confidence interval [CI]: 0.675–0.909; sensitivity: 59.46%; specificity: 91.67%), 0.705 (95% CI: 0.557–0.827; sensitivity: 51.35%; specificity: 91.67%), and 0.836 (95% CI: 0.702–0.926; sensitivity: 78.38%; specificity: 83.33%), respectively. This result indicated that the integration of αT-catenin and plakophilin-2 immunoreactivities can be used as a discriminator of molecular assays.

### 3.4. Loss of Co-Localization of ICD and αT-Catenin in ARVC

Furthermore, we analyzed the expression of desmosomal proteins in the ICD structure. IHC staining of the ICD shows that more than half of the autopsied ARVC patients have lost localization of αT-catenin, desmocollin-2, desmoglein-2, desmoplakin, plakoglobin, and plakophilin-2 compared with the controls (Figure 1d). The aforementioned findings were consistent with those observed in samples with and without pathogenic mutations (Table 3).

### 3.5. Validation of IHC Findings to Facilitate the Diagnosis of Clinical Patients with ARVC

Ten clinical specimens from 10 patients, including 5 with definite ARVC and 5 with structurally normal heart, who underwent endomyocardial biopsy were collected for validation. Table 4 demonstrated detailed information of these patients. The immunoreactivities of αT-catenin and plakophilin-2 significantly decreased in the ARVC group compared with the non-ARVC group (*p* = 0.028, Figure 2a,b).

### 3.6. Reduced Expression of Plakophilin-2 in CTNNA3 Knockdown Cells

We assessed the expression pattern of ICD-associated proteins using a cell model. Using siRNA, CTNNA3 gene expression was transiently knocked down to investigate the function of αT-catenin in mouse cardiomyocyte HL-1 cells. Western blotting was performed 48 h after siRNA transfection to determine protein levels. Figure 3a demonstrated the efficiency of siRNA silencing. Figure 3b showed that the expression of αT-catenin was suppressed by approximately 50.03%.

Then, we examined whether other ICD-associated molecules were affected by the silencing of CTNNA3. Notably, silencing CTNNA3 decreased the expression of desmocollin-2, plakophilin-2, and connexin 43 (86.70% ± 3.60%, 75.08% ± 11.58%, and 57.73% ± 18.62%; Figure 3b). Depletion of αT-catenin also led to an increase in the expression of transforming growth factor-beta 3 (117.50% ± 32.75%), which is an important marker of fibrogenesis (Figure 3a). The quantitative data from IHC and Western blotting showed similar protein expression trends in CTNNA3 mutant hearts and cells with CTNNA3 knockdown (Figure 3b,c).

### 3.7. Correlation between αT-Catenin and Plakophilin-2 Proteins

In light of the fact that depletion of αT-catenin reduces plakophilin-2 expression, it would be useful to identify the regulatory mechanism involving αT-catenin and plakophilin-2. Firstly, the proteins were knocked down using a lentiviral vector-based shRNA system, and it was observed that the knockdown of CTNNA3 and PKP2 did not affect each other’s mRNA expression (Figure 4a), but protein expressions were decreased (Figure 4b). Following MG101 treatment, the reduced protein was recovered (Figure 3b,c). As a consequence of post-translational proteolytic processing by cysteine proteases, αT-catenin and plakophilin-2 affected the stability of each other’s proteins.

## 4. Discussion

### 4.1. Major Findings

In the present study, IHC staining of autopsied patients with ARVC showed significant reductions in the expression of αT-catenin and plakophilin-2. These findings may provide crucial information that could assist in the accurate diagnosis of ARVC. A reduction in the expression of αT-catenin and plakophilin-2 has consistently been observed in clinical patients with definite ARVC who have undergone endomyocardial biopsy. Furthermore, after gene knockdown, αT-catenin and plakophilin-2 affect each other’s protein stability through post-translational proteolytic processing.

### 4.2. αT-Catenin and Plakophilin-2 in ARVC Disease

The CTNNA3 gene encoding αT-catenin in human cells has been mapped to chromosome 10q21.3, which belongs to the αT-catenin family [21]. In addition to interactions with the cytoskeleton, αT-catenin is a cytoplasmic molecule that is vital for the dynamic maintenance of tissue morphogenesis by integrating with the cadherin-catenin complex. The cytoplasmic domain of E-cadherin binds to the armadillo β-catenin and plakoglobin, which in turn binds α-catenin. In mammals, the α-catenin composite consists of three subtypes, namely αE-catenin, αN-catenin, and αT-catenin [29]. Among these subtypes, αT-catenin was found in cardiac tissues [15], and it interacts with N-cadherins and desmosomal cadherins, particularly plakophilin-2 [19]. In addition, the current study supports the correlation mentioned above. As a result of knocking down CTNNA3, plakophilin-2 expression decreased, mimicking the pathogenesis of ARVC. It is noteworthy that PKP2 is one of the most important gene mutations in ARVC [19]; this mutation is associated with electrical conduction defects and ventricular arrhythmias via voltage-gated sodium channels (Nav1.5) and Connexin 43 [30]. The reduced expression of CTNNA3 may lead to a decreased expression of plakophilin-2 and connexin 43, ventricular conduction disturbance, ventricular arrhythmogenesis, and sudden cardiac death in autopsied patients. Further, αT-catenin, desmocollin-2, desmoglein-2, desmoplakin, plakoglobin, and plakophilin-2 lost their localization to the ICD in ARVC (Figure 1d and Table 3). To the best of our knowledge, this study is the first to demonstrate that αT-catenin and plakophilin-2 are important in ARVC and that their decrease is characteristic of the disease. Furthermore, our study proved that the expression of both αT-catenin and plakophilin-2 by IHC could be the surrogate markers for the diagnosis of ARVC.

### 4.3. Decreases in αT-Catenin and Plakophilin-2 Can Facilitate the Diagnosis of ARVC

The role of plakoglobin staining in the diagnosis of ARVC has been reported previously with a sensitivity of 76–91% and a specificity of 57–84% [17,31,32,33,34]. Further, some studies demonstrated that IHC analysis of plakoglobin alone cannot serve as diagnostic criteria in patients with ARVC, especially from a clinical perspective [18,35,36]. Contrary to previous findings, the present study found that the level of plakoglobin expression in RV specimens from autopsied individuals with a diagnosis of ARVC was not significantly different from that of controls. The discrepancy between our results and those reported in previous studies may be explained by differences in disease severity, nonuniform genetic characteristics, and different ethnicities in the samples compared. Notably, the quantification of IHC data (Figure 1b) into a heatmap showed that the expression of αT-catenin was generally reduced in patients with ARVC, and that of plakophilin-2 and desmin partly decreased in patients with ARVC. Furthermore, similar patterns of protein expression were observed (Figure 2b and Figure 3a–c) after comparing the quantitative data between western blot analysis (CTNNA3 silencing cells) and immunohistochemistry (patients with CTNNA3 mutations). As a result of the present findings, the integration of the immunoreactivities of αT-catenin and plakophilin-2 could facilitate the diagnosis of ARVC.

### 4.4. Limitations

The current study had several limitations. First, ARVC exhibits a wide spectrum of clinical manifestations. Selection bias could confound the present results. Therefore, a large cohort should be included to validate the generalizability of the present findings. Second, the expression of αT-catenin decreased significantly in autopsied patients with ARVC, which was not consistent in clinical patients with ARVC. The abovementioned findings highlighted that the combination of αT-catenin and plakophilin-2 expression should be used to facilitate the clinical diagnosis, particularly during the early stage. Further investigation should be conducted to validate the role of αT-catenin and plakophilin-2 expression in a large cohort. Third, the clinical phenotype was not associated with the expression of αT-catenin and plakophilin-2 in the current study. Whether the expression of ICD-associated proteins is associated with specific clinical manifestations needs to be elucidated. Lastly, the disease entities of specimens obtained from clinical endomyocardial biopsy could be varied. Therefore, further studies are warranted to assess the applicability of using the expression of αT-catenin and plakophilin-2 to differentiate patients with non-ischemic cardiomyopathy rather than ARVC.

## 5. Conclusions

The depletion of αT-catenin reduced the expression of desmocollin-2, plakophilin-2, and connexin 43, thereby mimicking the pathogenesis of ARVC. However, αT-catenin and plakophilin-2 affect each other’s protein stability through post-translational proteolytic processing. The reduction and dislocalization of αT-catenin and plakophilin-2 on the ICD can be observed in autopsied patients and clinical patients with ARVC irrespective of the identifiable gene mutations. Therefore, it provides a diagnostic value using IHC staining to detect αT-catenin and plakophilin-2 to facilitate the diagnosis of ARVC.

## Figures and Tables

**Figure 1 ijms-23-05529-f001:**
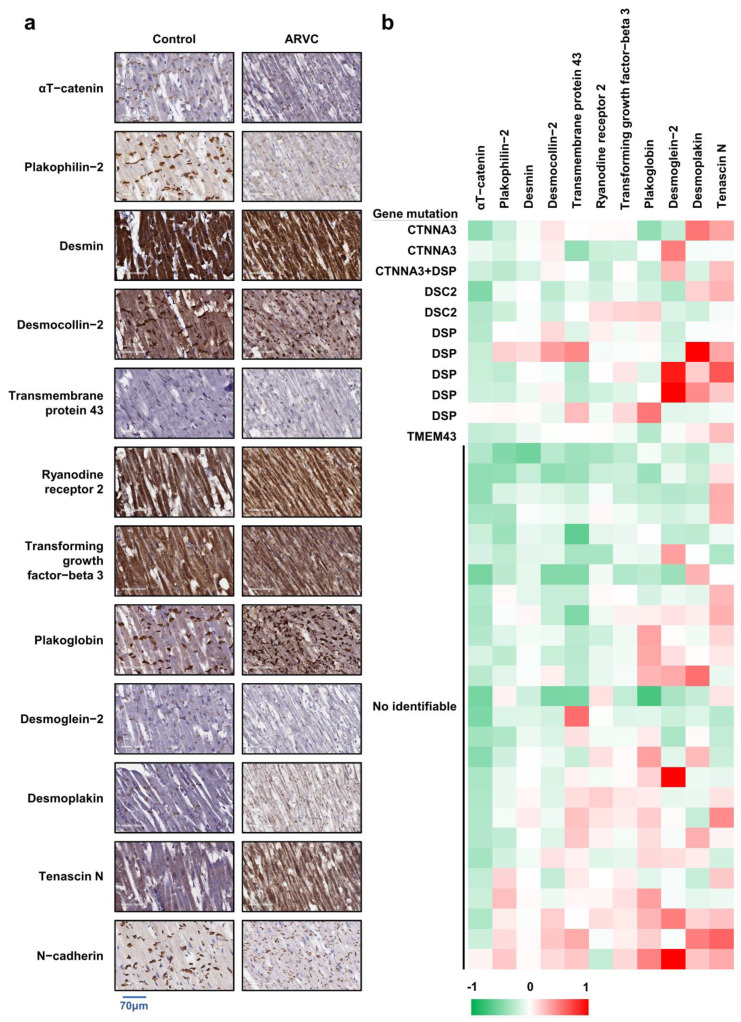
Representative immunohistochemical images of myocardial tissue from controls and ARVCs with different gene mutations. (**a**) The right ventricle obtained from the autopsy was assessed via IHC staining using the indicated antibodies. The immunoreactive intensity was evaluated using ImageJ, and the results are presented in Table 1. (**b**) Heatmap of protein expression in ARVC. The immunoreactive intensity was assessed using ImageJ, and the delta value was calculated to form a heatmap. Positive delta values (red) indicated that the expression of proteins was higher in patients with ARVC than those in controls, whereas a negative delta value (green) showed that the expression of proteins was lower in patients with ARVC than in controls. (**c**) Receiver operating characteristic curve for the prediction of ARVC based on the levels of αT-catenin and plakophilin-2, as measured on IHC staining. (**d**) Expression of desmosomal proteins in the ICD structure. Results showed one representative control case and ARVC cases with different gene mutations.

**Figure 2 ijms-23-05529-f002:**
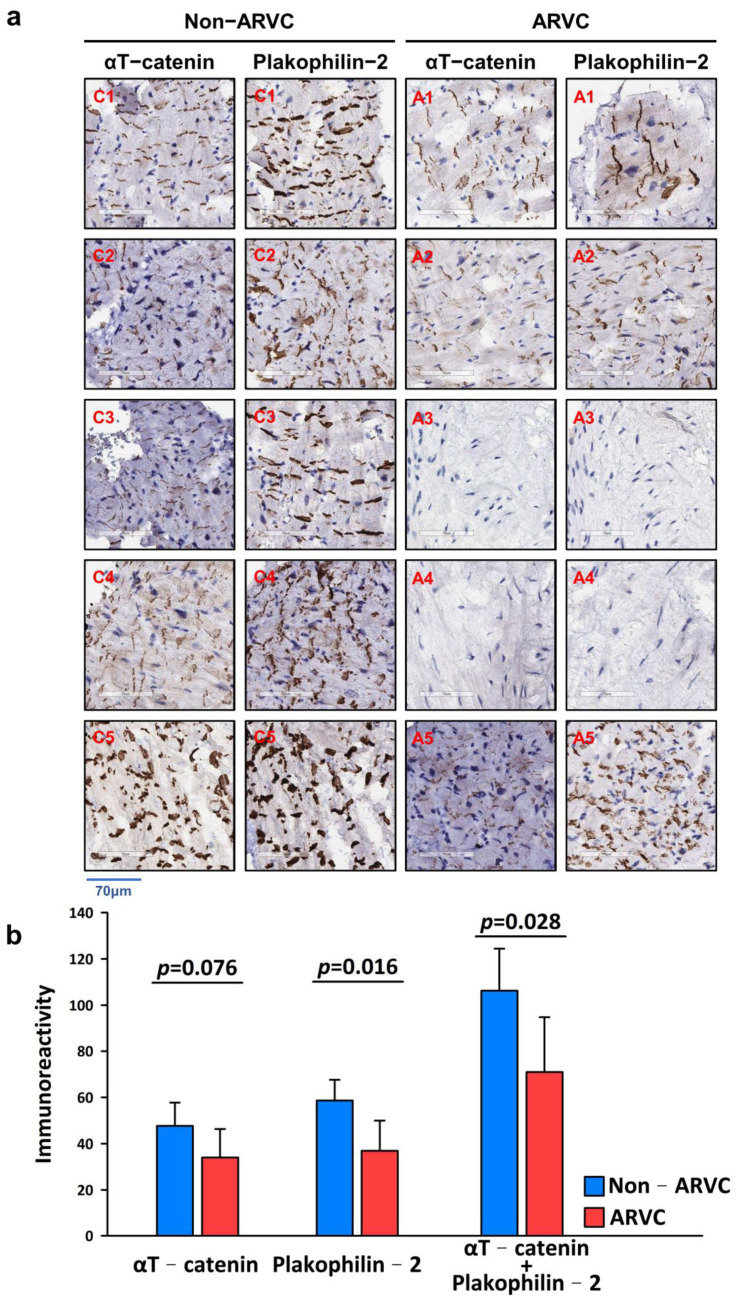
Representatives of myocardial immunohistochemistry images of specimens from clinical endomyocardial biopsies. (**a**) IHC staining of right ventricular tissues collected via endomyocardial biopsy was performed using αT-catenin and plakophilin-2 antibodies. (**b**) The immunoreactive intensity was assessed using ImageJ.

**Figure 3 ijms-23-05529-f003:**
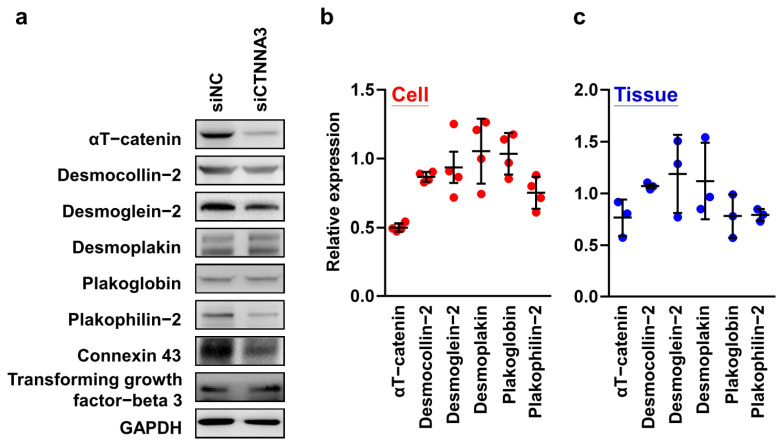
Analyses of protein expression in cardiomyocytes. (**a**) Protein levels in non-targeting control siRNA (siNC) and siCTNNA3 transfected HL-1 cardiomyocytes. Western blotting was performed with the indicated antibodies. Glyceraldehyde-3-phosphate dehydrogenase (GAPDH) was used as an internal control. The experiment was performed in quadruplicate. (**b**) Relative expression levels of HL-1 cardiomyocytes were calculated using the VisionWorks 8.20 software. The expression level was calibrated using GAPDH (internal control). (**c**) The immunoreactive intensity of IHC staining in patients with CTNNA3 mutations was analyzed using ImageJ. The relative intensity was calibrated in the control group.

**Figure 4 ijms-23-05529-f004:**
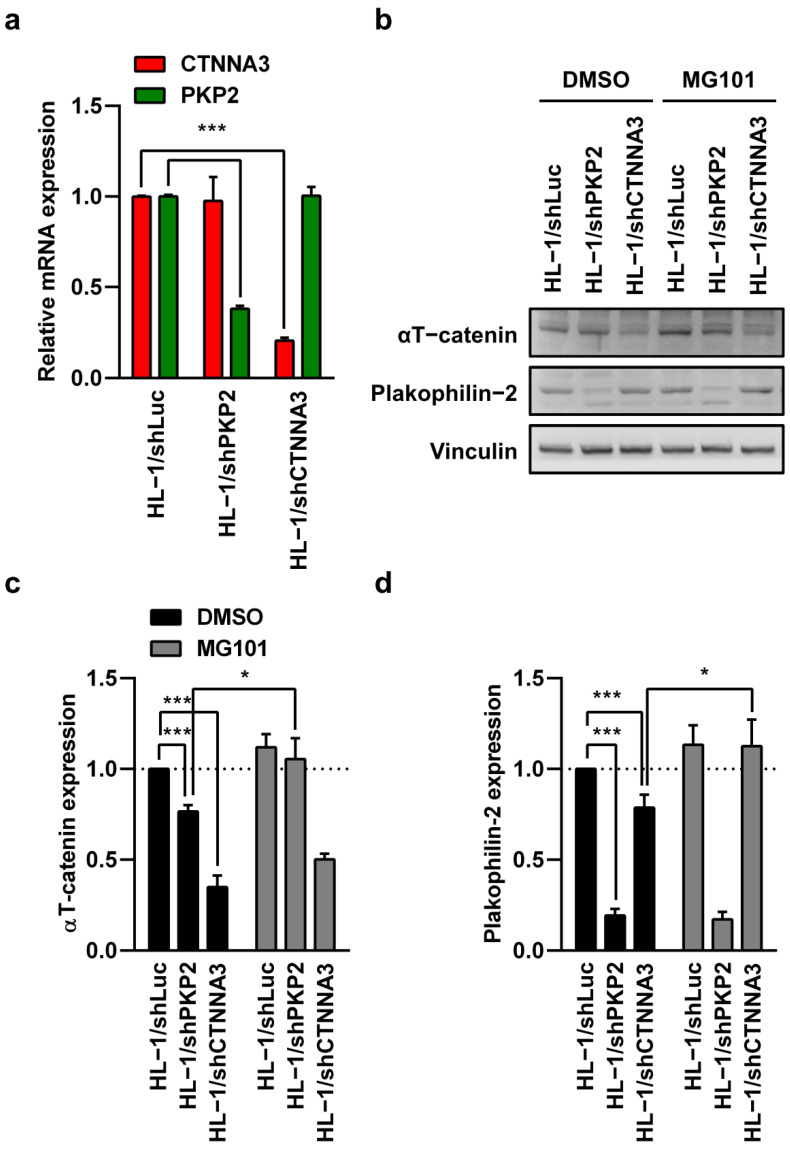
Knockdown of CTNNA3 and PKP2 affected each other’s protein levels, but not each other’s mRNA expression. (**a**) The mRNA levels of HL-1/shLuc, shCTNNA3, and shPKP2 stable cell lines were measured by real-time PCR using Power SYBR Green PCR Master Mix. Relative expression of CTNNA3 and PKP2 mRNA was normalized by β-actin. This graph was from three independent experiments. (**b**) Protein levels in HL-1/shLuc, shCTNNA3, and shPKP2 cardiomyocytes after MG101 treatment. Western blots were performed using the indicated antibodies. Vinculin was used as an internal control. The experiment was performed in quadruplicate. Relative protein expression levels were presented in the (**c**,**d**). * *p* < 0.05; *** *p* < 0.001.

**Table 1 ijms-23-05529-t001:** Parameters of RV specimens collected from autopsied patients.

Parameters	Controls	Patients with ARVC	*p* Value
Number of patients	12	37	-
Sex (female/male)	4/8	7/30	0.427
Age	44.17 ± 16.45	32.86 ± 10.30	0.021
Plakophilin-2 expression	71.00 ± 10.84	62.48 ± 12.74	0.034
Desmoplakin expression	64.58 ± 17.04	68.64 ± 19.17	0.471
Desmocollin-2 expression	118.38 ± 15.01	108.97 ± 22.45	0.114
Desmoglein-2 expression	49.69 ± 8.96	54.55 ± 23.11	0.780
Plakoglobin expression	103.44 ± 35.51	105.97 ± 27.92	0.972
αT-catenin expression	71.63 ± 20.56	50.95 ± 9.66	<0.001
Desmin expression	166.89 ± 12.11	158.62 ± 18.48	0.087
Ryanodine receptor 2 expression	150.81 ± 10.23	143.48 ± 19.65	0.259
Transmembrane protein 43 expression	116.38 ± 23.91	106.53 ± 32.93	0.245
Transforming growth factor-beta 3 expression	147.57 ± 13.81	143.35 ± 17.37	0.530
Tenascin N expression	111.85 ± 21.79	125.07 ± 26.52	0.178

**Table 2 ijms-23-05529-t002:** Comparison of immunoreactivity between autopsied ARVC patients with and without identifiable gene mutations.

Parameters	Patients with Identifiable Mutation (*n* = 11)	Patients without Identifiable Mutations (*n* = 26)	*p* Value
Age	32.45 ± 11.48	33.04 ± 10.00	0.523
Plakophilin-2 expression	62.54 ± 9.97	62.45 ± 13.92	0.935
Desmoplakin expression	76.86 ± 25.26	65.15 ± 15.21	0.160
Desmocollin-2 expression	121.02 ± 20.05	103.87 ± 21.77	0.036
Desmoglein-2 expression	58.57 ± 27.80	52.84 ± 21.21	0.909
Plakoglobin expression	99.26 ± 26.66	108.80 ± 28.46	0.181
αT-catenin expression	54.73 ± 10.00	49.36 ± 9.24	0.060
Desmin expression	163.91 ± 11.60	156.39 ± 20.50	0.517
Ryanodine receptor 2 expression	142.09 ± 18.22	144.06 ± 20.55	0.857
Transmembrane protein 43 expression	111.35 ± 28.87	104.49 ± 34.84	0.806
Transforming growth factor-beta 3 expression	147.25 ± 16.12	141.70 ± 17.91	0.441
Tenascin N expression	134.95 ± 25.01	120.90 ± 26.49	0.124

**Table 3 ijms-23-05529-t003:** Numbers of patients with reduced ICD protein expression between the ARVC group and the control group.

Patient Number	Patients with Identifiable Mutation (*n* = 11)	Patients without Identifiable Mutations (*n* = 26)	*p* Value
αT-catenin expression	7 (63.64%)	18 (69.23%)	1
Desmocollin-2 expression	8 (72.73%)	11 (42.31%)	0.151
Desmoglein-2 expression	9 (81.82%)	16 (61.54%)	0.279
Desmoplakin expression	7 (63.64%)	10 (38.46%)	0.279
Plakoglobin expression	1 (9.09%)	1 (3.85%)	0.512
Plakophilin-2 expression	7 (63.64%)	14 (53.85%)	0.723

**Table 4 ijms-23-05529-t004:** Baseline characteristics and fulfilled Task Force Criteria of clinical patients with and without definite ARVC for validation of IHC staining.

No.	Diagnosis	Age	Sex	Structure	Depolarization	Repolarization	Family History	VA	Gene Mutations	Tissue Biopsy
A1	ARVC	45	Female Male	Major	Minor	Major	(+)	Major	DSG2	Minor
A2	ARVC	50	Male	Major	Minor	Major	(-)	Major	(-)	Minor
A3	ARVC	34	Male	Major	Minor	Major	(-)	Major	PKP2	Minor
A4	ARVC	35	Male	Major	Major	Major	(-)	Major	DSG2	Minor
A5	ARVC	16	Female	Minor	Minor	Major	(+)	Major	(-)	(-)
C1	Idiopathic VT	66	Male							
C2	NICM *	74	Female							
C3	Idiopathic VF	61	Male							
C4	NICM *	64	Male							
C5	NICM *	31	Male							

* Nonischemic cardiomyopathy rather than ARVC. NICM: nonischemic cardiomyopathy; VA, ventricular arrhythmia; VT, ventricular tachycardia; VF, ventricular fibrillation.

## Data Availability

Not applicable.

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
