# Peer review of "Decreased Expression of Plakophilin-2 and αT-Catenin in Arrhythmogenic Right Ventricular Cardiomyopathy: Potential Markers for Diagnosis"

_ijms, 2022, doi:10.3390/ijms23105529_

Round 1

Reviewer 1 Report

Dear Authors,

I congratulate you for the interesting article. I have some minor suggestions.

Abstract - You should be more specific about the purpose of the study.

Introduction - well written.

The Materials and Methods section should be the second Section.

Results - 

  • Figure 1 should be moved after the section of the text that mentions it
  • For Figure 1c, the results are described in the text; it does not need to be written in the legend of the figure also
  • Line 152 and 166 with references 19 and 21 should be moved to the Introduction chapter
  • Figure 3- I think it should be split into two or more, for a better understanding

Discussion - Well written

Reviewer 2 Report

This manuscript studies the expression potential biomarkers plakophilin-2 and αT-catenin in cardiac disease. The data presented is clear and easy to understand. I believe the findings are clinically relevant and are supported by proper results. However, discussion can be improved. 

Overall, manuscript is good and will attract wide readership. 
